# Time Series Modeling at Scale: A Universal Representation Across Tasks and Domains

## Abstract

Time series are ubiquitous, capturing real-world phenomena ranging from human neuronal firing and tectonic activity to atmospheric conditions. However, they are challenging to analyze due to domain-specific timescales (*e.g.*, sub-second for brain activity and years for weather phenomena), complex multivariate relations, and disparate modeling objectives. Prior works model time series by targeting specific tasks, like forecasting, or distinct domains, like neural recordings. We introduce a universal approach for scalable time series modeling across many tasks and domains, which we call TOTEM: **To**kenized **T**ime Series **Em**beddings. We propose a task-agnostic embedding that projects a continuous time series of any length onto a discrete set of learned tokens. This embedding is derived by optimizing a self-supervised objective formulated as a task-independent convolution-based vector quantized variational autoencoder. Drawing inspiration from the recent successes of Large Language Models, these discrete token sequences are then used to learn downstream models with the powerful Transformer architecture. We show that TOTEM matches or achieves SOTA performance on forecasting, classification, and translation tasks with data drawn from a myriad of domains: neuroscience, seismology, meteorology, power grids, and urban traffic. We further demonstrate TOTEM's scalability by introducing and evaluating it on new datasets, the largest being $\sim 14\times$ larger than existing benchmarks. Finally, we illustrate TOTEM's dominant zero-shot generalization capabilities across all of our downstream tasks.

## 1 Introduction

Time series capture the dynamics of diverse real-world systems over a range of timescales and levels of granularity. The huge spectrum of intricate patterns, variation in temporal horizon, and diverse research objectives makes general time series modeling challenging. For example, consider the distinct nature of three domain-task pairs: long-term weather forecasting, human brain machine interface classification, and seismological sensor translation. Because these domains and tasks are so disparate, many works consider specific tasks, like forecasting (Das et al., 2023; Challu et al., 2023; Zeng et al., 2023; Nie et al., 2022; Zhou et al., 2022; 2021; Wu et al., 2021; Liu et al., 2021; Li et al., 2019), or domains, like neural recordings (Peterson et al., 2022; Liu et al., 2022; Talukder et al., 2022; Peterson et al., 2021; Lawhern et al., 2018). Moving towards more universal modeling approaches has the potential to unlock unprecedented generalized performance across a range of tasks and domains at scale, akin to recent breakthroughs in language modeling.

In this work, we develop a scalable and universal approach for time series modeling across many diverse domain-task pairs without domain- or task-specific data preprocessing and feature engineering. We draw inspiration from Large Language Models (Radford et al., 2018), which leverage universal tokenizations to solve a variety of tasks via Transformers (Vaswani et al., 2017). However, because language data is naturally discrete while time series are continuous, one may wonder whether time series can be effectively discretely tokenized to enable task-agnostic modeling. To that end, our contributions are as follows.

**Contributions to Scale and Universality.**

- We study time series modeling at **scale**, which we interpret in three ways: (1) number of tasks, (2) number of domains, and (3) dataset size. In this work, we explore all three

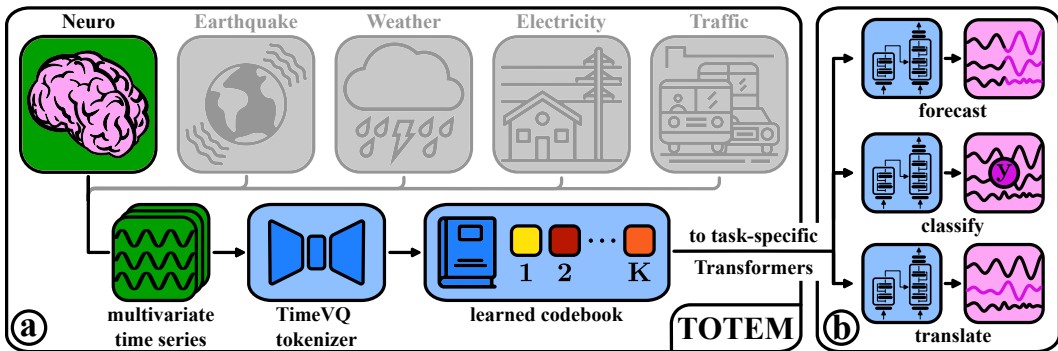

Figure 1: **(a)** TOTEM employs TimeVQ, a VQ-VAE based model, to tokenize multivariate time series from any domain, including neuroscience, seismography, meteorology, power grids and urban traffic. **(b)** A single learnt codebook is the foundational embedding for distinct downstream tasks.

by evaluating on three tasks (forecasting, classification, and translation) and five domains (termed Neuro, Earthquake, Weather, Electricity, and Traffic). Moreover, we introduce new datasets for analysis that are up to $\sim 14\times$ larger than previous benchmarks.

- We also study the **universality** of our representations, which we interpret in two ways: (1) their adaptability across tasks, and (2) their generalizability to out-of-distribution data from the same domain. We show that our method adapts to many tasks without domain-specific data engineering and that it achieves strong zero-shot generalization capabilities.

**Technical Contributions.**

- Using a VQ-VAE based tokenizer, called TimeVQ, we show that discretely tokenizing time series is an effective and task-agnostic modeling approach.

- We show that using our tokenizations, simple Transformer-based submodules can solve diverse time series modeling tasks at state of the art levels without relearning the embeddings.

- We integrate TimeVQ and these Transformers into a single method called **TOTEM** (**To**kenized **T**ime Series **Em**beddings) and show that it matches or beats the best existing methods designed for specific domains and/or tasks without further engineering.

## 2   RELATED WORK

We draw from work on time series analysis and tokenization in Large Language Models.

**Forecasting.** Traditional time series forecasting methods, *e.g.*, ARIMA and GARCH (Box & Jenkins, 1968), are effective for short-horizon predictions but make linearity assumptions not applicable to complex datasets and long horizons. Recently, multi-layer neural networks have emerged as powerful alternatives. Here, some works (Nie et al., 2022; Zhou et al., 2022; 2021; Wu et al., 2021; Liu et al., 2021; Li et al., 2019) advocate for Transformers and others (Das et al., 2023; Challu et al., 2023; Zeng et al., 2023) for linear models or MLPs. While our proposed tokenization scheme is compatible with any such downstream model architecture, we elect to use Transformers due to their strong empirical performance on sequenced data. The current state of the art methods in time series forecasting are TiDE (Das et al., 2023) and PatchTST (Nie et al., 2022). TiDE uses an MLP-based encoder that leverages covariates, such as the day of the week or holidays, as effective features. PatchTST adopts a Transformer encoder architecture and represents time series data with patches that can be overlapping or non-overlapping. We compare against them in our experiments.

**Classification.** Multivariate time series classification (Ruiz et al., 2021; Karim et al., 2019; Bagnall et al., 2018) has been explored with a focus on neuroscience applications (Liu et al., 2022; Peterson et al., 2022; 2021; Ye & Pandarinath, 2021; Lawhern et al., 2018). Electrophysiological neuroscience datasets record the brain with electrodes whose physical dimensions range from micrometers (resulting in single neuron recordings) to millimeters (resulting in local field potential based electroencephalography and electrocorticography recordings). This 1000-fold difference in electrode size results in vastly different statistical properties between single neuron and local field

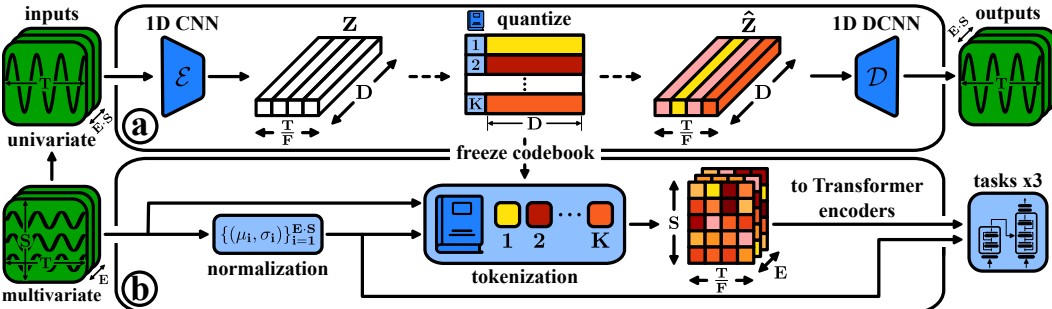

Figure 2: **(a)** TimeVQ learns a codebook from univariate time series via autoencoding. The encoder and decoder consist of 1D convolutions and transpose convolutions, respectively. The input encoding $z$ is quantized into $\hat{z}$ via the codebook prior to decoding. **(b)** All sensors are independently normalized before tokenization. The token sequences are consumed by task-specific downstream Transformers.

potential (LFP) datasets. Therefore, neuroscience modeling techniques typically only validate their results on datasets from one data regime, *i.e.*, single neuron (Liu et al., 2022; Ye & Pandarinath, 2021) or LFP-based (Peterson et al., 2021; Lawhern et al., 2018). Recently, Peterson et al. (2022; 2021) released a large electrocorticography (ECoG) data suite intended for brain machine interface classification. TOTEM is evaluated against the popular CNN-based EEGNet (Lawhern et al., 2018) used for LFP modeling and forms the backbone of other brain data classifers (Peterson et al., 2021).

**Translation.** In multivariate time series, imputation of missing data is commonly studied (Talukder et al., 2022; Yıldız et al., 2022; Luo et al., 2019; 2018). Typically, the missing data independently afflict multiple sensors at random times or at all sensors at the same times, so many models leverage data in and across channels to impute missing values. We study an even harder task that we call translation: the inference of completely unobserved sensor channels from observed ones. Translation is key in many applications which suffer from catastrophic hardware failures or sensor corruption.

**Tokenization.** In language modeling, Byte-Pair Encoding (Gage, 1994) extracts the most common contiguous character sequences via a recursive algorithm. Language models like GPT (Radford et al., 2018) use this representation to learn task-agnostic token embeddings via self-supervised next-token prediction, followed by supervised finetuning for downstream tasks. In image reconstruction and generation, VQ-VAEs (Van Den Oord et al., 2017) and VQ-GANs (Esser et al., 2021; Rombach et al., 2022) quantize images into discrete learnt codes. Other models, such as ViTs (Dosovitskiy et al., 2020) and PatchTST (Nie et al., 2022), patchify their input data to be compatible with Transformers, essentially learning continuous embeddings of input time series. We opt for a discrete tokenization scheme, similar to LLMs, which generates a fixed set of tokens used for learning downstream models without the need for retraining. This contributes to the scalability of our approach and allows zero-shot generalization to out-of-distribution settings.

## 3 TOTEM: TOKENIZED TIME SERIES EMBEDDINGS

This section presents TOTEM, our universally applicable method for scalable time series modeling. TOTEM comprises two distinct modules: TimeVQ and Downstream Transformers. The TimeVQ module adopts a vector quantized variational autoencoder (VQ-VAE) (Van Den Oord et al., 2017) to learn a set of tokens. The tokens form a discrete codebook learnt from continuous time series via self-supervision. We use the codebook's embeddings as the basis for task-specific Downstream Transformers. We describe TimeVQ in § 3.1 and our Downstream Transformer models in § 3.2. Each multivariate time series dataset consists of $E$ examples (*i.e.*, number of distinct recordings), $S$ sensor channels, and $T$ time steps, and can be formally expressed as $\{\mathbf{x}_j\}_{j=1}^{E} \subset \mathbb{R}^{S \times T}$.

### 3.1 TIMEVQ: TOTEM'S TASK-AGNOSTIC TOKENIZER

VQ-VAEs (Van Den Oord et al., 2017), introduced for image reconstruction, are designed to quantize an input representation into discrete codes. Here, we introduce TimeVQ, a vector quantization-based model that analogously aims to learn a codebook of discrete time series tokens, which we hypothesize is an effective tokenization even though time series are naturally continuous phenomena.

The codebook is learned via self-supervision and can be thought of as a waveform basis that best reconstructs the input time series. Figure 2 shows an overview of TimeVQ and Figure 3 visualizes learnt codebooks for the Neuro domain (other domains in Appendix).

**Model.** TimeVQ consists of an encoder, quantizer, latent codebook, and decoder. It takes in a univariate time series $\{\mathbf{x}_i \in \mathbb{R}^T\}_{i=1}^{E \cdot S}$ obtained by flattening the sensor channel of the multivariate data. Thus, TimeVQ is sensor-agnostic, enabling TOTEM's zero-shot generalizability (see § 4.6).

TimeVQ's encoder $\mathcal{E}$ consists of strided 1D convolutions compressing the time series by a cumulative stride of $F$. $\mathcal{E}$ maps a univariate time series $\mathbf{x} \in \mathbb{R}^T$ to a latent representation $\mathbf{z} = \mathcal{E}(\mathbf{x}) \in \mathbb{R}^{T/F \times D}$, where $D$ is the the hidden dimension. The latent codebook $\mathcal{C} = \{\mathbf{c}_i\}_{i=1}^K$ consists of $K$ $D$-dim codewords $\mathbf{c}_i \in \mathbb{R}^D$. During quantization, the codebook is used to replace $\mathbf{z}$ with $\hat{\mathbf{z}} \in \mathbb{R}^{T/F \times D}$ such that $\hat{\mathbf{z}}_j = \mathbf{c}_k$, where $k = \arg\min_i ||\mathbf{z}_j - c_i||_2$. Our decoder $\mathcal{D}$ follows the reverse architecture of the encoder $\mathcal{E}$, consisting of 1D transpose convolutions with a cumulative stride of $1/F$ mapping the quantized $\hat{\mathbf{z}}$ to a reconstructed time series $\hat{\mathbf{x}} = \mathcal{D}(\hat{\mathbf{z}}) \in \mathbb{R}^T$.

**Training Objective.** We learn $\mathcal{E}$, $\mathcal{D}$, and $\mathcal{C}$ by optimizing the objective $\mathcal{L} = \mathcal{L}_{\text{rec}} + \mathcal{L}_{\text{cmt}}$ consisting of a reconstruction loss $\mathcal{L}_{\text{rec}} = \frac{1}{E \cdot S} \sum_i ||\mathbf{x}_i - \hat{\mathbf{x}}_i||_2^2$ and a commitment loss $\mathcal{L}_{\text{cmt}}$, identical to Van Den Oord et al. (2017), which allows the codebook to update despite the the non-differentiable $\arg\min$ operation during quantization. TimeVQ is agnostic to any downstream task, as it is only trained to reconstruct the input time series. This makes our representation scalable: for any time series domain, *e.g.* neural firing, we learn a single codebook that can be used for any downstream application. Put simply, TOTEM's task-agnostic TimeVQ becomes the foundational embedding for any downstream model. We empirically show the scalable and universal properties of our representation in § 4, where we employ a single TimeVQ for three distinct tasks.

**Waveform vs. Scale.** In time series data, both the scale (*i.e.*, absolute values) and the waveform (*i.e.*, shape) are important characteristics. Since time series can, in principle, have any scale (even within a domain), learning a sufficiently expressive basis of waveforms across all scales is intractable. Thus, prior to tokenization, we normalize the data by subtracting the mean $\mu_i$ and dividing by standard deviation $\sigma_i$ for each time series $\mathbf{x}_i$. This allows our tokens to be scale-invariant, capturing only information about the shape of the data in a given domain. However, because the scale is important for interpreting the data, we reintroduce it when training downstream task-specific models.

## 3.2 Downstream Transformers: TOTEM's Task-Specific Time Series Models

We demonstrate how to utilize a TimeVQ task-agnostic codebook for three tasks: forecasting, classification and translation. As described in § 3.1, TimeVQ transforms a continuous real-valued univariate time series of length $T$ into a sequence of $T/F$ discrete tokens, where $F$ is the cumulative stride, or compression factor, of the model. This token sequence is aptly suited for the powerful Transformer architecture (Vaswani et al., 2017), which learns relationships between tokens by applying attention mechanisms in the embedding space. Unlike in prior popular architectures (*e.g.* RNN's inherent sequence linearity or CNN's spatial locality), Transformers lack inductive biases, thereby enabling more expressive models in language and vision.

When adapting Transformers to multivariate time series tasks, we aim to create distinct and reusable submodules that can be mixed and matched to solve any downstream task. In total, we instantiate three: (1) the Time Transformer Encoder, **Time XEncoder**; (2) the Sensor Transformer Encoder, **Sensor XEncoder**; and (3) the **Scale Reinstater**. Our design allows us to construct models for our trio of tasks – forecasting, classification and translation – using our reusable submodules, as we describe below. We emphasize that while the submodules are combined in slightly different ways for each task due to factors like the shape of the data, the expected outputs, etc., the submodule architectures themselves are fixed beforehand and designed in a task-agnostic way.

**Forecasting.** In forecasting, our predictive model utilizes the **Time XEncoder** and **Scale Reinstater** submodules. Using the codebook, TimeVQ converts a sensor's observed measurements $\mathbf{x}_s \in \mathbb{R}^{T_{\text{in}}}$ to a sequence of $T_{\text{in}}/F$ tokens. Time XEncoder processes these tokenized time series independently for each sensor, adding time-based positional encodings to each token along the time dimension. Using a series of multi-head attention layers, Time XEncoder predicts the forecasted measurements $\bar{\mathbf{y}}_s \in \mathbb{R}^{T_{\text{out}}}$ for $s = 1, ..., S$. In parallel, the Scale Reinstater module (realized as an MLP) takes in $\mathbf{x}_s$

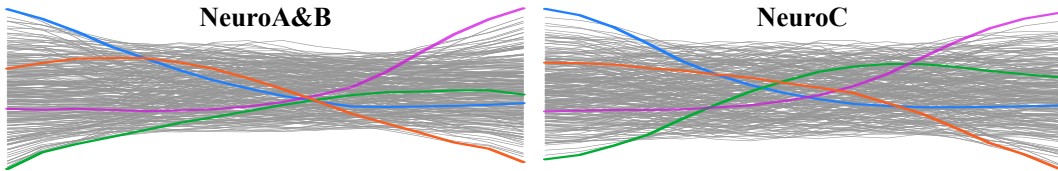

Figure 3: We visualize all 256 tokens in the *time domain* highlighting some for better visualization. When comparing NeuroA&B and NeuroC codebooks we see similar codebook distributions, allowing us to visualize how strong zero-shot generalization is possible. More visualizations in the Appendix.

and predicts the future's mean, $\mu_s$, and standard deviation, $\sigma_s$, for each sensor $s = 1, ..., S$. The final forecasted prediction is $\mathbf{y}_s = \sigma_s \cdot \bar{\mathbf{y}}_s + \mu_s$. Both Time XEncoder and Scale Reinstater are trained in a supervised fashion by minimizing three smooth L1 losses between predictions $\{\bar{\mathbf{y}}_s, \mu_s, \sigma_s\}$ and their ground truth respectively.

**Classification.** In classification, we predict a label $\mathbf{y}$ for every example $\mathbf{x} \in \mathbb{R}^{S \times T}$. Thus, our classifier must holistically model interactions across the sensor and time dimensions. To this end, it consists of three submodules: the **Time XEncoder**, the **Sensor XEncoder** and the **Scale Reinstater**. As in forecasting, the Time XEncoder first produces tokenized time series with time-based positional encodings. The last layer averages the outputs into a single $D$-dim vector for each sensor, which we call a sensor summarization token. The sequence of $S$ summarization tokens is then passed to the Sensor XEncoder, where a positional encoding corresponding to a sensor's position is added. The classifier's Scale Reinstater linearly projects each sensor's scale statistics $[\mu_s, \sigma_s]$ to a $D$-dim encoding which is added to the corresponding sensor summarization token. Finally, the Sensor XEncoder outputs categorical probabilities. All models are jointly trained by optimizing a cross-entropy loss.

**Translation.** Here, the goal is to recover unobserved sensor readings, $\mathbf{y} \in \mathbb{R}^{S_{\text{out}} \times T}$, from observed ones $\mathbf{x} \in \mathbb{R}^{S_{\text{in}} \times T}$, which can be thought of as imputing completely missing sensors. The translation model consists of a **Time XEncoder**, a **Sensor XEncoder** and a **Scale Reinstater**. For each of the $S_{\text{in}}$ available sensors, the Time XEncoder outputs a sequence of $T/F$ tokens with added time-based positional encodings. These are flattened into $(\frac{T}{F} \cdot S_{\text{in}})$ tokens and input to the Sensor XEncoder. The Scale Reinstater, identical to the one in classification, adds a sensor scale encoding to the tokens. All models are trained with a smooth L1 loss between predicted and true translation measurements.

For all aforementioned tasks, we apply our downstream models to in-distribution and out-of-distribution settings. In time series modeling, zero-shot generalization to out-of-distribution time series is very challenging due to the high variability of time series measurements.

## 4 EXPERIMENTS

We evaluate TOTEM on three distinct tasks, forecasting, classification, and translation; and five distinct domains, human brain LFP (Neuro); phase-sensitive seismograms (Earthquake); meteorological indicators (weather); domestic electricity consumption (Electricity); and road occupancy rate (Traffic). Through extensive experimental evaluations, we show that TOTEM achieves two main results. First, models trained with TOTEM match or outperform prior task- and/or domain-specific methods even without additional engineering. Second, our experiments on zero-shot generalization demonstrate that TOTEM's learned tokenizations produce models that generalize on out-of-distribution data within the same domain (i.e., from an unseen dataset), surpassing baseline performance across all domains.

### 4.1 DATASETS

Among our five domains, we introduce two new ones (Neuro and Earthquake) to study TOTEM's generalizability. For **forecasting**, we use Weather, Electricity, and Traffic data sourced from popular time series forecasting benchmarks (Wu et al., 2021). In addition, we incorporate three human-brain ECoG datasets (Peterson et al., 2022) which have never been paired with a long-horizon forecasting task. These datasets correspond to different patients, which we label A, B, and C. For the 96 time step prediction task, Traffic contains roughly $17k$ examples, Electricity $26k$, Weather $52k$, NeuroA $143k$, NeuroB $157k$, and NeuroC $710k$. NeuroA & B are roughly $3\times$ and NeuroC is roughly $14\times$ the

| MAE (↓) | Codebook Size $K$ | | | Code Dim $D$ | | Compression $F$ | |
|---|---|---|---|---|---|---|---|
| | 32 | 256 | 512 | 64 | 256 | 4 | 16 |
| NeuroB | 0.1160 | **0.0729** | 0.0749 | **0.0729** | 0.0880 | **0.0729** | 0.1966 |
| Weather | 0.1049 | 0.0674 | **0.0638** | **0.0674** | 0.0740 | **0.0674** | 0.1418 |
| Earthquake | 0.0018 | **0.0015** | 0.0017 | **0.0015** | 0.0018 | **0.0015** | 0.0023 |
| MSE (↓) | | | | | | | |
| NeuroB | 0.0249 | **0.0101** | 0.0106 | **0.0101** | 0.0151 | **0.0101** | 0.0712 |
| Weather | 0.0391 | 0.0162 | **0.0131** | **0.0162** | 0.0228 | **0.0162** | 0.0696 |
| Earthquake | 0.0003 | **0.0002** | 0.0002 | **0.0002** | 0.0002 | **0.0002** | 0.0008 |

Table 1: **TimeVQ study.** Reconstruction performance for various codebook sizes, codeword dimensions, and compression factors.

| Neuro | C → C | A & B → C |
|---|---|---|
| MAE (↓) | 0.0821 | 0.0839 |
| MSE (↓) | 0.0119 | 0.0129 |
| Earthquake | Random Split | Geo Split |
| MAE (↓) | 0.0015 | 0.0111 |
| MSE (↓) | 0.0002 | 0.0007 |

Table 2: **TimeVQ zero-shot.** Reconstruction performance of TimeVQ models trained in distribution (2nd column) and out-of-distribution (3rd column).

size of the previous largest benchmark. For **classification**, we utilize NeuroA, B, and C as no other datasets come with classification labels. Finally, for **translation**, we introduce a dataset of phase sensitive seismograms (Sun et al., 2023; Zhu & Beroza, 2019) in addition to NeuroA, B, and C.

## 4.2 TIMEVQ EFFECTIVENESS

We first study three attributes of the codebook: the number of codes $K$, the code dimension $D$, and the compression factor $F$ (*i.e.*, the stride of TimeVQ's encoder $\mathcal{E}$). We report the reconstruction error on three domains, NeuroB, Weather and Earthquake in Table 1. A codebook size of 32 performs worse across all domains, with 256 and 512 performing equally well. A codeword dimension of 64 works as well or better than 256. Lastly, a compression factor of 4 performs better than 16. This is expected as TimeVQ tokens trained with lower compression factors each capture fewer time steps. For all experiments, we use $K = 256$, $D = 64$, and $F = 4$.

Figure 3 visualizes all learnt codes in the time domain (decoded by TimeVQ's decoder $\mathcal{D}$) for Neuro data (see more domains in the Appendix). We visualize all codes in gray and highlight a few codes with color, allowing visualization of the codes' shape diversity between domains.

Table 2 shows TimeVQ's reconstruction error for in- and out-of-distribution, *i.e.* zero-shot, evaluations. We explore TimeVQs that are (1) trained and evaluated on NeuroC; (2) trained on NeuroA&B and evaluated on NeuroC; (3) trained and evaluated on all Earthquake data; and (4) trained on NorCal Earthquake and evaluated on SoCal Earthquake (called Geographical Split). Experiments 1 and 3 evaluate performance on in-distribution held-out data, while 2 and 4 test zero-shot generalization on out-of-distribution data. Overall, the zero-shot models achieve only slightly worse performance, which supports our hypothesis that TOTEM learns generalized embeddings (see Table 2). We study the application of these zero-shot TimeVQs to downstream tasks in § 4.6.

## 4.3 FORECASTING

For forecasting, we evaluate on Weather, Electricity, and Traffic from Wu et al. (2021) as well as NeuroA, B, and C – a total of six datasets. We compare TOTEM to the two best approaches from the past two years, PatchTST (Nie et al., 2022) and TiDE (Das et al., 2023).

**Task & Metrics.** In forecasting, models intake a time series $\mathbf{x} \in \mathbb{R}^{S \times T_{\text{in}}}$ and predict future readings $\mathbf{y} \in \mathbb{R}^{S \times T_{\text{out}}}$, where $S$ is the number of sensors and $T_{\text{in}}, T_{\text{out}}$ signify the durations of the preceding and succeeding time series, respectively. The pairs $(\mathbf{x}, \mathbf{y})$ are generated by striding the original time series data. For our task, we experiment with forecast windows $T_{\text{out}} \in \{96, 192, 336, 720\}$. The input time window, commonly referred to as lookback, is set to $T_{\text{in}} = 512$, following PatchTST (Nie et al., 2022). TiDE reports a lookback of $T_{\text{in}} = 720$ for Weather, Electricity and Traffic, which gives them an advantage. We take the Weather, Electricity, and Traffic results for TiDE and PatchTST from the TiDE paper. For the Neuro domain, we train TiDE and PatchTST using their publicly available code, and set the lookback to $T_{\text{in}} = 512$ for all methods for a fair comparison. NeuroA, B, and C originally only record 1001 continuous timesteps. Therefore, with a lookback window of 512, the data only support lookahead lengths of $T_{\text{out}} \in \{96, 192, 336\}$. We follow prior work and report mean squared error (MSE) and mean absolute error (MAE) between ground truth and predicted future readings.

**Training Details.** We emphasize that for all prediction lengths, we only train *one* TimeVQ per dataset. The downstream model is a Time XEncoder with 4 layers and 4 attention heads and a feed-forward

| Dataset (Size) | NeuroC (710k) | | | NeuroB (157k) | | | NeuroA (143k) | | | Weather (52k) | | | | Electricity (26k) | | | | Traffic (17k) | | | |
|---|---|---|---|---|---|---|---|---|---|---|---|---|---|---|---|---|---|---|---|---|---|
| MAE (↓) | 96 | 192 | 336 | 96 | 192 | 336 | 96 | 192 | 336 | 96 | 192 | 336 | 720 | 96 | 192 | 336 | 720 | 96 | 192 | 336 | 720 |
| TOTEM (ours) | **0.585** | **0.611** | **0.622** | **0.600** | **0.642** | **0.663** | **0.360** | **0.385** | **0.399** | **0.196** | 0.242 | 0.283 | **0.330** | 0.231 | 0.245 | 0.265 | 0.292 | **0.241** | **0.242** | **0.248** | 0.275 |
| TiDE | 0.597 | 0.616 | 0.625 | 0.608 | 0.644 | 0.664 | 0.365 | 0.388 | 0.402 | 0.222 | 0.263 | 0.301 | 0.340 | 0.229 | 0.243 | 0.261 | 0.294 | 0.253 | 0.257 | 0.260 | **0.273** |
| PatchTST | 0.597 | 0.616 | 0.625 | 0.626 | 0.655 | 0.669 | 0.374 | 0.392 | 0.402 | 0.198 | **0.241** | **0.282** | 0.334 | **0.222** | **0.240** | **0.259** | **0.290** | 0.249 | 0.256 | 0.264 | 0.286 |
| MSE (↓) | | | | | | | | | | | | | | | | | | | | | |
| TOTEM (ours) | **0.647** | **0.710** | **0.740** | **0.674** | **0.779** | 0.832 | **0.260** | **0.305** | **0.331** | **0.147** | 0.195 | 0.248 | 0.314 | 0.135 | 0.151 | 0.168 | 0.200 | 0.369 | 0.383 | 0.397 | 0.446 |
| TiDE | 0.670 | 0.719 | 0.742 | 0.692 | 0.781 | **0.829** | 0.267 | 0.307 | **0.331** | 0.166 | 0.209 | 0.254 | **0.313** | 0.132 | 0.147 | 0.161 | 0.196 | **0.336** | **0.346** | **0.355** | **0.386** |
| PatchTST | 0.671 | 0.720 | 0.743 | 0.737 | 0.807 | 0.845 | 0.287 | 0.319 | 0.337 | 0.149 | **0.194** | **0.245** | 0.314 | **0.129** | **0.147** | **0.163** | **0.197** | 0.360 | 0.379 | 0.392 | 0.432 |

Table 3: **Forecasting.** We compare TOTEM to TiDE and PatchTST on 6 datasets, arranged in decreasing size, and prediction horizons $T_{\text{out}} = \{96, 192, 336, 720\}$. We report mean absolute error (MAE) and mean squared error (MSE). For neuro, $T_{\text{out}} \leq 336$ because of its original size.

| ACC (↑) | NeuroC | NeuroB | NeuroA |
|---|---|---|---|
| TOTEM (Ours) | **0.736** | **0.633** | **0.599** |
| EEGNet | 0.542 | 0.488 | 0.538 |
| Majority Class | 0.530 | 0.506 | 0.518 |

| | E & N → Z | | Z → E & N | | NeuroC | | NeuroB | | NeuroA | |
|---|---|---|---|---|---|---|---|---|---|---|
| | MAE | MSE | MAE | MSE | MAE | MSE | MAE | MSE | MAE | MSE |
| TOTEM (Ours) | **0.491** | **0.644** | **0.479** | **0.719** | 0.534 | **0.435** | **0.504** | **0.390** | **0.517** | **0.416** |
| Baseline | 0.499 | 0.883 | 0.504 | 0.977 | **0.530** | 0.461 | 0.518 | 0.434 | 0.518 | 0.440 |

Table 4: **Classification** on Neuro.    Table 5: **Envelope Translation** on Earthquake and Neuro.

hidden dimension of 256. We train using Adam with a base learning rate of 0.0001 and a one cycle learning rate scheduler in accordance with PatchTST.

**Results.** Table 3 shows forecasting performance for NeuroA, B, and C, alongside Weather, Electricity, and Traffic. The datasets are arranged in descending order by size. If we calculate the number of wins, ties, and losses for each model across all 21 experiments for the MAE metric, TOTEM wins $^{14}/_{21}$, TiDE wins $^{1}/_{21}$, and PatchTST wins $^{6}/_{21}$. For the MSE metric, TOTEM wins $^{8}/_{21}$, TiDE wins $^{9}/_{21}$, and PatchTST wins $^{3}/_{21}$. Notably for the newly introduced $14\times$ and $3\times$ datasets; in MAE TOTEM wins $^{9}/_{9}$, TiDE wins $^{0}/_{9}$, and PatchTST wins $^{0}/_{9}$; in MSE TOTEM wins $^{7}/_{9}$, TiDE wins $^{1}/_{9}$, and PatchTST wins $^{0}/_{9}$. These results demonstrate that TOTEM starkly outperforms prior approaches for larger dataset sizes. Overall, TOTEM performs best; see Figure 4 for a visualization of a NeuroA forecast over a length-96 horizon.

## 4.4 Classification

For classification, we experiment on NeuroA, B, and C, which contain labels of human activity associated with brain machine interface signals. The task is to predict the *move* or *rest* label from human ECoG measurements. We compare our method to EEGNet (Lawhern et al., 2018).

**Task & Metrics.** In this task, the multivariate time series input $\mathbf{x} \in \mathbb{R}^{S \times T}$ is mapped to the action executed by the patient, denoted by $\mathbf{y} \in \{\text{move}, \text{rest}\}$. We follow the train/val/test split in the original data suite release (Peterson et al., 2022), where the last recording day is used as the test set. We report accuracy, namely the percentage of correct action predictions, on the test set.

**Training Details.** We use a trained TimeVQ to tokenize the respective Neuro datasets. In the downstream model, both Time and Sensor XEncoder consist of 2 layers and 2 attention heads. The feed-forward hidden dimension of each layer is set to 128. We train using Adam, a base learning rate of 0.0001 and a one cycle learning rate scheduler.

**Results.** Table 4 shows action classification performance on NeuroA, B, and C. On these datasets, TOTEM significantly outperforms EEGNet by 6.1, 14.5, and 19.4 percentage points respectively. EEGNet utilizes 2D convolutions to operate in the sensor-time space, effectively treating the multi-variate time series like an image; therefore, the electrode position is assumed to be fixed for every input. In contrast, TOTEM leverages a sensor-agnostic TimeVQ and an order-agnostic transformer architecture, a design decision which allows us to learn tokens on data with one arrangement of sensors and deploy them on data with another. For brain recordings, this is crucial, as each patient has a unique arrangement of surgically implanted arrays, so model generalization is highly desirable. We explore this further in our zero-shot experiments in § 4.6.

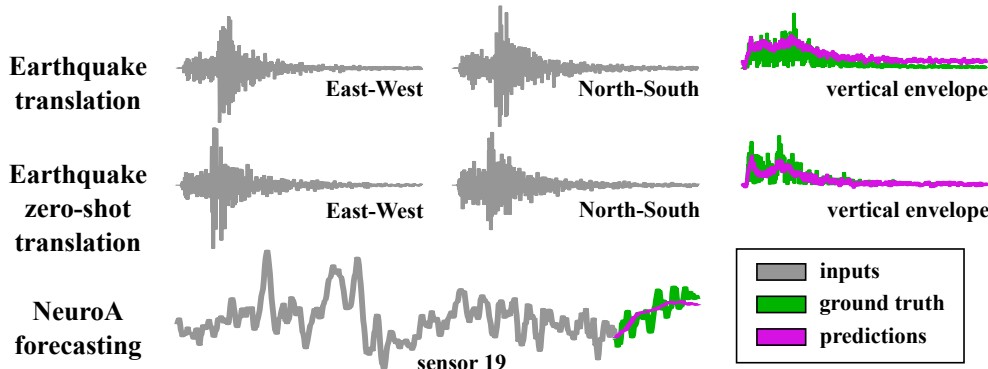

Figure 4: Model predictions for translation and forecasting. We show the inputs, predicted and true measurements. Our model is able to predict the vertical envelope in both the in-domain and zero-shot setting with similar accuracy. In all examples our models predict denoised versions of the true data, indicating that our tokenization scheme might effectively denoise multivariate time series data.

## 4.5 TRANSLATION

In the translation task, we use measurements from observed sensors to infer missing measurements from unobserved sensors (*i.e.*, full sensor imputation). Formally, given input $\mathbf{x} \in \mathbb{R}^{S_{\text{in}} \times T}$, we predict output $\mathbf{y} \in \mathbb{R}^{S_{\text{out}} \times T}$, where $S_{\text{in}}, S_{\text{out}}$ are the numbers of observed and predicted sensors respectively.

**Task & Metrics.** We evaluate translation performance on the Earthquake and Neuro datasets. In seismology, sensors measure three axes of movement: east-west (E), north-south (N), and vertical (Z). The E and N components mainly record the secondary seismic wave, while the Z component mainly measures the primary seismic wave. When sensor corruption occurs, one or more of the {E, N, Z} sensor readings can be lost, making translation between sensor waveforms invaluable, especially in the context of earthquake early-warning systems. In neuroscience, sensor imputation is critical as fixing failed electrodes requires invasive brain surgery.

Due to the inherent difficulty of imputing an entire missing sensor, we relax the translation task by instead predicting the envelopes of the missing waveforms, defined as their Hilbert transforms (Oppenheim, 1999). While easier to predict, the envelope is still useful, as it contains amplitude and power information, critical to earthquake early warning systems and neural analysis.

In the Earthquake domain, we conduct experiments on two envelope translation variants, from E&N to Z measurements (E & N → Z) and vice versa (Z → E & N). To evaluate prediction quality, we report mean squared error (MSE) and mean absolute error (MAE) between predicted and ground truth measurements. We visualize some examples in Figure 4. For the Neuro translation task we randomly mask 90% of sensors and train the model to return the envelope of these missing sensors for $T = 192$, which is the most challenging masking ratio from Talukder et al. (2022).

**Training Details.** We use trained TimeVQs to tokenize the Neuro and Earthquake datasets. In the downstream models, both Time and Sensor xEncoders consist of 4 layers and 4 attention heads. The feed-forward hidden dimension of each layer is set to 256. We train using Adam, a base learning rate of 0.0001 and a one cycle learning rate scheduler.

**Results.** Since we introduce the sensor translation task, there are no readily available competing methods. Therefore, we design an MLP baseline that intakes time measurements $\mathbf{x}$ and predicts envelopes $\mathbf{y}$. Table 5 shows translation performance for the two Earthquake experiments and the sensor masking experiments on NeuroA, B, and C. TOTEM outperforms the MLP baseline in all experiments on the MSE metric and in $^4/_5$ experiments on the MAE metric.

## 4.6 ZERO-SHOT GENERALIZATION

**Setup.** In this section, we apply the previously discussed zero-shot TimeVQs (Table 2) to explore within-domain zero-shot generalization (*i.e.*, on unseen datasets with different data distributions) across all tasks. Zero-shot generalization is exceedingly difficult due to the high variability within

**(a) Forecasting.**

| MAE (↓) | Neuro A & B → C | | |
|---|---|---|---|
| | 96 | 192 | 336 |
| TOTEM (Ours) | **0.599** | **0.618** | **0.627** |
| TiDE | 0.616 | 0.630 | 0.637 |
| PatchTST | 0.612 | 0.627 | 0.636 |
| MSE (↓) | | | |
| TOTEM (Ours) | **0.674** | **0.726** | **0.749** |
| TiDE | 0.708 | 0.743 | 0.764 |
| PatchTST | 0.709 | 0.746 | 0.767 |

**(b) Classification.**

| Neuro A & B → C | ACC (↑) |
|---|---|
| TOTEM (Ours) | **0.569** |
| EEGNet | 0.475 |
| Majority Class | 0.530 |

**(c) Translation.**

| NorCal E&N → SoCal Z | MAE (↓) | MSE (↓) |
|---|---|---|
| TOTEM (Ours) | **0.506** | **0.590** |
| Baseline | 0.562 | 0.914 |
| NorCal Z → SoCal E&N | | |
| TOTEM (Ours) | **0.496** | **0.624** |
| Baseline | 0.581 | 1.047 |

Table 6: Zero-shot generalization results for downstream tasks.

domains, *e.g.*, fault lines in different areas of the world behave differently. In all cases, both the TimeVQs and Downstream Transformers never have access to the held out entity. In Neuro, we hold out dataset NeuroC, while in Earthquake, we hold out SoCal Earthquakes. In Neuro, different patients have both differing numbers of electrodes (NeuroA: 106, NeuroB: 72, Neuro: 93) and differing sensor locations on the brain. In Earthquake, NorCal and SoCal are composed of distinct fault lines that produce remarkably different waveforms due to factors like varying material subsurface properties. For zero-shot forecasting and classification, we utilize the Neuro domain, and for zero-shot translation we utilize the Earthquake domain.

**TimeVQ Generalizability.** The zero-shot performance of TimeVQ can be interpreted through the lens of Figure 3. There, we visualize all 256 (decoded) codebook tokens for TimeVQ trained on data from patients A & B and on patient C, noting that the overall shape of the aggregated codes is nearly identical. In particular, the highlighted codes show that similar rising and falling behaviors are always captured from brain recordings regardless of patient, demonstrating the universality of the learned tokenization. These qualitative features provide insight into the strong zero-shot generalization on Neuro data.

**Downstream Transformer Generalizability.** In Table 6, we explore zero-shot forecasting with NeuroA, B, and C. Across all prediction horizon lengths and on both MSE and MAE metrics, TOTEM measurably outperforms TiDE and PatchTST. We similarly explore zero-shot classification performance on all Neuro datasets, where we find that TOTEM significantly outperforms EEGNet. Finally, on Earthquake data, we test zero-shot generalizability on the E& N → Z and Z → E& N translation tasks. In both cases, we significantly outperform the baseline on all metrics, which demonstrates the value of our discrete tokenization scheme. Overall, we observe that TOTEM consistently outperforms all existing state of the art models on every single zero-shot generalization task, which provides strong evidence that discrete tokenization enables general, scalable time series modeling.

# 5 CONCLUSIONS, LIMITATIONS, & FUTURE WORK

We present TOTEM, a universal methodology for time series analysis at scale. TOTEM is comprised of TimeVQ and Downstream Transformer modules that process TimeVQ's output tokens for forecasting, classification, and translation. TOTEM demonstrates strong zero-shot performance and state of the art performance across multiple tasks and domains.

**Limitations & Future Work.** (1) TimeVQ's convolutional architecture can tokenize a univariate time series of any length $T$. However, when $T$ isn't a multiple of the compression factor, $F$, some time steps may be omitted from tokenization. At high compression factors, this could result in many time steps being excluded from downstream tasks. (2) TimeVQ's token length is determined by a fixed compression factor $F$. However, considering the continuous nature of time series, a dynamic token length might offer a more fitting representation. Introducing variability in token lengths has the potential to enhance data representations, possibly boosting performance in downstream tasks. (3) We only explored a VQ-VAE based tokenizer. Incorporating other methods from the field of Neural Compression into TOTEM could be an intriguing direction. (4) TOTEM introduces a universal approach to time series analysis at scale, maintaining a consistent methodology across various tasks and domains. Yet, each domain still necessitates its own TimeVQ, and every task demands a distinct downstream model. The eventual goal in task-agnostic domain-agnostic time series analysis is to have a singular TimeVQ tokenizer for all domains and one core architecture across all tasks, essentially creating a time series foundation model. This is a topic for future exploration.

## 6 CODE OF ETHICS

There are no immediate ethical concerns that arise from our work. However, as with all data driven methods, certain ethical risks are important to be discussed, in this case surrounding time series modeling. A few are reported below:

**Privacy Concerns.** Time series data, especially when sourced from personal devices or applications, can contain sensitive information about individuals, *e.g.* for health domains. In this work, no time series were sourced from personal devices. And the neural data we use was de-identified by the original dataset authors.

**Misuse.** Time series forecast models can be misused. For instance, if a model forecasts stock prices or market movements, it could be exploited for insider trading or other illegal financial activities. In this work, we are focused on domains pertinent to scientific disciplines.

**Economic Impacts.** Automated forecasts and decisions based on time series models can significantly impact industries and labor markets both positively and negatively. For instance, if a model can accurately predict weather patterns, it might affect farmers and their crop decisions, or if it can forecast energy consumption, it could impact the energy sector.

## 7 REPRODUCIBILITY

We address the reproducibility of our approach on three axis: data availability, data processing and model training & evaluation.

**Data Availability.** All data used in this work are publicly available and referenced. See § 4.1.

**Data Processing.** The time series data in our work come in various forms, most commonly as a 3D array of shape $E \times S \times T$. Depending on the specific downstream tasks, we manipulate this data to conform to the required format of the respective task. For forecasting, we follow the striding implemented by Autoformer (Wu et al., 2021) in their publicly available github repository. For classification, the data is kept in its original form (Peterson et al., 2022), and we remove faulty sensors and examples (e.g. all zeros). For translation, we extract the different axes of movement from the released data (Zhu & Beroza, 2019), and remove faulty sensors and examples (e.g. all zeros). All data preprocessing scripts will be made available during our code and model release.

**Model Training & Evaluation.** We introduce a TimeVQ module which extends VQ-VAEs for the time series domain. We describe TimeVQ's design in § 3.1. For a compression factor of $F = 4$, TimeVQ's encoder $\mathcal{E}$ consists of 2 conv layers, each followed by a relu. TimeVQ's decoder $\mathcal{D}$ inverts this design and consists of 2 deconv layers each followed by a relu. We train TimeVQ with Adam, a base learning rate of $0.0001$, and input time series of length 96 for $15000$ iterations at a $4096$ batchsize across all datasets and tasks. The Downstream Transformer modules are described in § 3.2 and their architecture and training details for each task are provided in the respective sections in epxeriments, § 4. We evaluate our time series predictions by reporting three metrics: accuracy, `MAE` and `MSE`. Model training, evaluation and pre-trained models will be released with our code release.

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

# A   APPENDIX

We provide more visualizations, dataset information and results in the Appendix.

## A.1   NEURO STATISTICS

Table 7 details the statistics on NeuroA,B,C. We show the dataset sizes in their original form as well as after processing for the downstream tasks. For the task of forecasting, NeuroC approximately hold $710k$ examples, which is roughly $14\times$ larger than previous benchmarks.

| Dataset | NeuroC | NeuroB | NeuroA |
|---|---|---|---|
| Original | $(1802, 93, 1001)$ | $(399, 72, 1001)$ | $(363, 106, 1001)$ |
| Forecast | $(709988, 93, 512)$ | $(157206, 72, 512)$ | $(143022, 106, 512)$ |
| Classify | $(1802, 93, 1001)$ | $(399, 72, 1001)$ | $(363, 106, 1001)$ |
| Translate | $(536996, 93, 192)$ | $(118902, 72, 192)$ | $(108174, 106, 192)$ |

Table 7: **Neuro Dataset Information.** All entries take the form $(E, S, T)$ where $E$ is the number of examples, $S$ is the number of sensors, $T$ is the number of time steps. The **Original** row details each neuro's initial shape before downstream task processing. The dataset is processed for the various tasks. For forecasting and translation, the data is strided similar to Wu et al. (2021). For classification, we keep the original shapes.

## A.2   CODEBOOK VISUALIZATION

Figure 5 visualizes codebooks for Neuro, Weather and Earthquake. For Neuro, we visualize the codebook from each individual patient as well as the codebook from the union of Neuro A and B time series. The latter codebook is used for zero-shot experimentation on NeuroC both for reporting reconstruction performance but also for downstream forecasting (see § 4.6). For Earthquake, we visualize the codebook for all earthquakes as well as the codebook for earthquakes in Northern California (NorCal).

## A.3   ADDITIONAL FORECASTING RESULTS

We show forecasting on the ETT benchmark from Wu et al. (2021). The ETT benchmark is an additional benchmark from the electricity domain which is processed to hold measurements every minute (ETTm) or every hour (ETTh). Table 8 compares TOTEM to TiDE and PatchTST and reports `MAE` and `MSE` for four output time horizons. We notice that TOTEM performs on par well on the larger ETTm, but worse on the significantly smaller ETTh. This result along with Table 3, where TOTEM dominates on larger datasers, suggests that our approach is more effective for learning at scale, *i.e.* with large scale datasets. For smaller datasets, MLP-based methods like TiDE seem to have an advantage.

| Dataset | ETTm1 | | | | ETTm2 | | | | ETTh1 | | | | ETTh2 | | | |
|---|---|---|---|---|---|---|---|---|---|---|---|---|---|---|---|---|
| MAE | 96 | 192 | 336 | 720 | 96 | 192 | 336 | 720 | 96 | 192 | 336 | 720 | 96 | 192 | 336 | 720 |
| TOTEM (ours) | 0.343 | 0.365 | 0.384 | 0.416 | 0.252 | 0.292 | 0.327 | 0.383 | 0.404 | 0.434 | 0.457 | 0.500 | 0.351 | 0.411 | 0.471 | 0.594 |
| Tide | 0.349 | 0.366 | 0.384 | 0.413 | 0.251 | 0.289 | 0.326 | 0.383 | 0.398 | 0.422 | 0.433 | 0.465 | 0.336 | 0.380 | 0.407 | 0 .451 |
| PatchTST | 0.346 | 0.370 | 0.392 | 0.420 | 0.256 | 0.296 | 0.329 | 0.385 | 0.400 | 0.429 | 0.440 | 0.468 | 0.337 | 0.382 | 0.384 | 0.422 |
| MSE | | | | | | | | | | | | | | | | |
| TOTEM (ours) | 0.294 | 0.334 | 0.366 | 0.425 | 0.166 | 0.227 | 0.281 | 0.364 | 0.379 | 0.427 | 0.455 | 0.503 | 0.285 | 0.367 | 0.450 | 0.651 |
| Tide | 0.306 | 0.335 | 0.364 | 0.413 | 0.161 | 0.215 | 0.267 | 0.352 | 0.375 | 0.412 | 0.435 | 0.454 | 0.270 | 0.332 | 0.360 | 0.419 |
| PatchTST | 0.293 | 0.333 | 0.369 | 0.416 | 0.166 | 0.223 | 0.274 | 0.362 | 0.370 | 0.413 | 0.422 | 0.447 | 0.274 | 0.341 | 0.329 | 0.379 |

Table 8: Forecasting Results on the ETT dataset

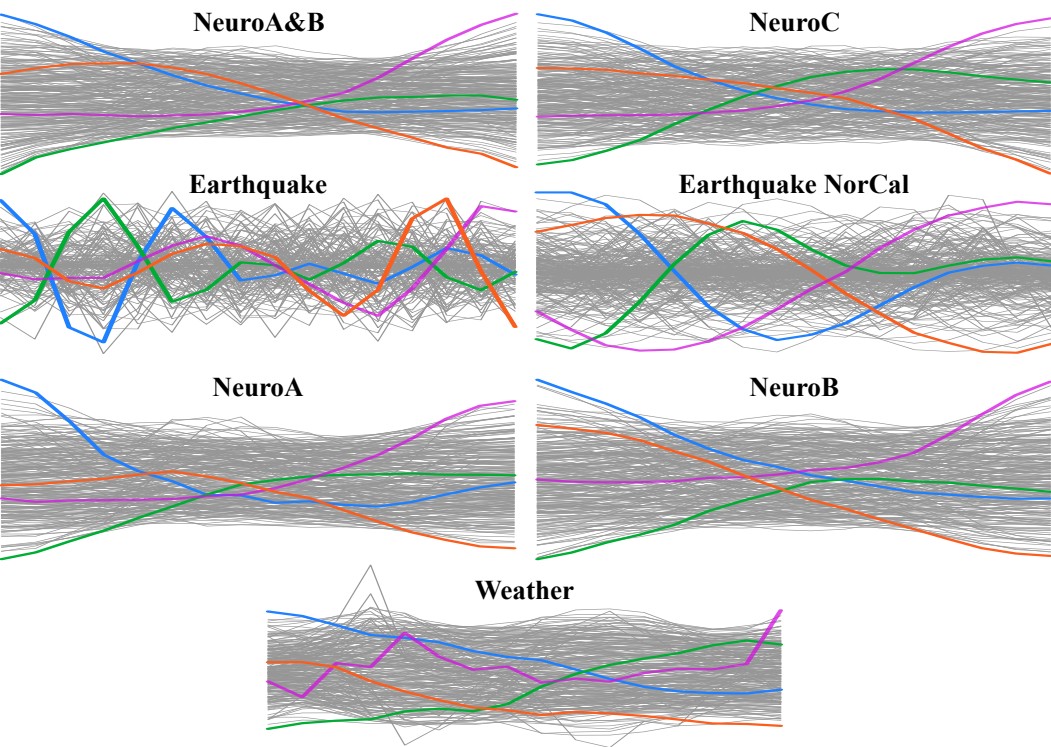

Figure 5: Codebooks in time space, visualizing all 256 codes for NeuroA, NeuroB, NeuroC, NeuroAB, Earthquake, Earthquake NorCal, and Weather. With selected codes visualized. Codebooks are quantitatively similar within domains.

