# OpenReview forum: "Time Series Modeling at Scale: A Universal Representation Across Tasks and Domains"
_ICLR.cc/2024/Conference — Submitted to ICLR 2024_

### Official Review · Reviewer_EFAQ · 2023-10-26

**Soundness:** 2 fair
**Presentation:** 1 poor
**Contribution:** 2 fair
**Rating:** 3
**Confidence:** 4

**Summary:**

This paper focuses on the universal representation learning of time series. By adopting the VQ-VAE, TOTEM is proposed to embed the input time series into several tokens. Afterwards, they employ the transformer encoder and task-specific heads for different tasks. Experimentally, the learned VQ-VAE embedding can generalize to different domains. The proposed method performs well in three different tasks.

**Strengths:**

1.	The paper focuses on an essential problem, that is learning generalizable representations for time series.
2.	Using VQ-VAE for embedding is interesting and reasonable. And the experiments verify that the learning embeddings generalize well in different domains.
3.	The authors experiment on several different tasks and domains.

**Weaknesses:**

1.	About the presentation and citation of the method section.

The model descriptions in Section 3.2 are ambiguous. For example, in Figure 1(b) and Figure 2, they plot the Transformer structure with both encoder and decoder, while in Section 3.2, they present the XEncoder as the “Time Transformer Encoder”. In my understanding, they only adopt the Transformer encoder part. More explanations are required. Besides, how to apply the Transformer encoder is also unclear? I think they use attention along the token dimension, but there are no clear descriptions here. I think they should present the model architecture in Figures, just like Figure 2.

The usages of Scale Reinstater are inconsistent in different tasks. In the forecasting task, the Scale Reinstater is used to provide future mean and standard deviation, while in classification and translation, they are used for scale embedding. I don’t think the usage in classification and translation can be named as “Reinstater”. A clearer way is expected for this section.

2.	I believe that three are several highly relevant works that are not properly cited.

In Section 3.1, they mention that TimeVQ receives the univariate time series by flattening the sensor channel. Flattening channels for univariate time series is firstly used in Crossformer [1].

In Section 3.2, they process the input series by removing mean and std and then recover the scale information by MLP, which is same as the usage in Non-stationary Transformer [2].

It is notable that I am not attempt to owning these designs in TOTEM to previous works, since they can be used for different proposes. However, these methods should be discussed at least.

In addition to the technical design, TimesNet [3] also focuses on the task-general backbone for time series, which should be discussed and compared.

[1] Crossformer: Transformer Utilizing Cross-Dimension Dependency for Multivariate Time Series Forecasting, ICLR 2023

[2] Non-stationary Transformers: Exploring the Stationarity in Time Series Forecasting, NeurIPS 2022.

[3] TimesNet: Temporal 2D-Variation Modeling for General Time Series Analysis, ICLR 2023

3.	About main experiments.

As I aforementioned, TimesNet is an important baseline, which is also evaluated on forecasting, classification and imputation tasks. They should compare with it in all three tasks.

Besides, the proposed model performs worse than other models on the Weather, Traffic and Electricity datasets.

4.	About the efficiency comparison.

Since they flatten the sensor channel of the multivariate data, they model efficiency will be extremely poor when it comes to time series with a lot of dimensions, such as Traffic. The efficiency comparison between TOTEM and other baselines is required, such as GPU memory, running time and parameter size.

5.	Ablation study is missing.

An important ablation is to replace the TimeVQ with one simple linear layer conducted on patch, which can support the motivation in utilizing VQ-VAE for time series encoding.

6.	The discussion about the translation and imputation tasks.

I can understand that translation is a challenging task. But imputation is also challenging since it requires the model to handle the missing data. It is also important to discuss the model performance in the missing data scenario.

7.	I think the title is kind of over-claimed.

Actually, TimesNet also adopts larger datasets. You can find them in the anomaly detection task in the TimesNet’s paper (more than 500k series). Thus, I think it is hard to claim “at scale”, since the dataset size is not that large.

Besides, they mentioned “across domains and tasks”. But only cross domain zero-shot experiments are conducted. For different tasks, they still need to retrain the model and task-specific heads.

**Questions:**

All the questions are listed above, including method description, experiment design and claims in title.

---

### Official Review · Reviewer_tudQ · 2023-10-30

**Soundness:** 3 good
**Presentation:** 2 fair
**Contribution:** 2 fair
**Rating:** 3
**Confidence:** 5

**Summary:**

The paper proposes a two-stage approach to time series modelling. The first stage learns a VQ-VAE model to learn a codebook to transform time series data into a sequence of discrete tokens (with continuous representations). The second stage feeds these tokens into downstream models (Transformers) to perform specific tasks. These tasks include forecasting, classification, and translation.

**Strengths:**

The paper tackles the ambitious problem of universal representations for time series. The paper is generally well written, having good visualisations and explanations of the proposed method.

**Weaknesses:**

1. Please consider some prior work using VQ-VAE similarly in vision [1] and in time series [2].
2. Claims made in the title and introduction are somewhat overstated. Claims of "scale" and "universality" leads one to think that a single model is learned across all datasets, but that is not true. Instead, one model is trained per dataset. What is the difference with existing paradigm such that this method can be called "universal"?
3. Claims that the newly introduced datasets are much larger than existing datasets are not well supported -- please include more details of these newly introduced datasets with exact details on how the reported values (first paragraph of section 4.1 and table 7) are derived.
4. Insufficient comparison to baselines. Specifically, see the line of work in [3] and many follow up works. They also tackle many different time series tasks.


[1] Bao, Hangbo, et al. "Beit: Bert pre-training of image transformers." arXiv preprint arXiv:2106.08254 (2021).

[2] Rasul, Kashif, et al. "VQ-AR: Vector Quantized Autoregressive Probabilistic Time Series Forecasting." arXiv preprint arXiv:2205.15894 (2022).

[3] Wu, Haixu, et al. "Timesnet: Temporal 2d-variation modeling for general time series analysis." arXiv preprint arXiv:2210.02186 (2022).

**Questions:**

1. Isn't the "translation" task just time series regression?
2. What are "envelopes" in the translation task"? Please give detailed mathematical formulation of this new task, it is not clear from the current writing.
3. In what situations do we want to perform forecasting on Neuro and Earthquake datasets? These do not seem like useful real world scenarios.

---

### Official Review · Reviewer_9M6r · 2023-10-31

**Soundness:** 2 fair
**Presentation:** 2 fair
**Contribution:** 2 fair
**Rating:** 3
**Confidence:** 5

**Summary:**

This paper introduces a pretraining approach designed to address the complexities of time-series data from diverse domains, tasks, and scenarios. The challenge at hand lies in bridging substantial disparities between various data sources, such as those within the realms of weather and traffic. Building upon the previously published foundations of the VQVAE algorithm, the method leverages a preexisting codebook, modifying it to suit a range of downstream tasks. The author posits that the resulting representation exhibits universality, allowing the learned codewords to seamlessly transfer across a spectrum of downstream datasets and domains.

**Strengths:**

1. The method is characterized by its simplicity and its clear and accessible presentation.

2. The literature review is commendable, effectively contextualizing the research within its broader academic framework.

3. The paper demonstrates a comprehensive approach to downstream tasks, encompassing forecasting, classification, and translation.

**Weaknesses:**

This work exhibits several weaknesses in its approach:

1. The method lacks a clear and intuitive motivation. Time-series data are inherently complex and challenging for human understanding, while tokens or word sequences are more easily comprehensible. Thus, the quantization of real-value time-series signals may not yield significant benefits.

2. The algorithm heavily relies on the VQVAE method, with only minor modifications applied to the original algorithm. Consequently, the level of novelty introduced by this work is relatively low.

3. The assertion of a "universal" representation appears overstated. While the results for Neuro datasets in forecasting show significant improvements over baselines, the performance on other datasets is relatively subpar, casting doubt on the universality claim. Furthermore, the statement that "we only train one TimeVQ per dataset" suggests that the learned representation is dataset-specific, which contradicts the concept of universality.

4. The experimental methodology is not robust. It is unclear whether the method was evaluated with multiple random runs, and the absence of standard deviation values for different experiment settings raises concerns about the reliability of the results.

5. The addition of new datasets seems arbitrary, as it primarily includes human-brain ECoG datasets. The motivation behind selecting these specific domain datasets over others remains unclear.

6. The exploration of domain gaps in the experiments is unsatisfactory. Training on data from one set of human domains and transferring to another, with small domain gaps, lacks persuasiveness. The absence of evaluations in more significant domain gap scenarios raises questions about the method's adaptability to varying data domains.

In summary, this paper appears to suffer from several key issues. Firstly, it lacks novelty in its methodology, as it heavily builds upon existing techniques with only minor modifications. Secondly, there is room for significant improvement in the experimental design and execution, including the need for more rigorous evaluation metrics and consideration of standard deviations. Finally, the paper is marred by over-claiming statements that may not be fully supported by the results presented. These aspects collectively contribute to a less convincing and impactful research contribution.

**Questions:**

1. Why do you use CNN structure in pretraining while utilizing Transformer model in downstream tasks?

2. Are the normalization methods different in Stage a and Stage b? According to Figure 2, the operations seem different in the figure.
What are the meanings of the arrows without any annotations in Stage b? For example, the arrow line above the block of "normalization", and the arrow line below the tokenization block.

3. What's the learning situations of different predictor such as the Scale Reinstater?

4. Did you evaluate the methods with different random runs? What are the std. values for different experiment settings?

---

### Meta-Review · Area_Chair_Ji9z · 2023-12-04

**Metareview:**

The manuscript was reviewed by three reviewers and all unanimously suggested rejection. Moreover, authors did not respond to the reviews. Following the reviews and authors lack of response, I also recommend rejection.

**Justification For Why Not Higher Score:**

All the raised issues by the reviewers are valid and authors chose to not respond.

**Justification For Why Not Lower Score:**

N/A

---

### Decision · Program_Chairs · 2024-01-16

Reject